# The Multiple Cooperative Mechanism and Globalization Path of Small Inland Cities in China: A Showcase Study of Dunhuang, China

**DOI:** 10.3390/ijerph191811241

**Published:** 2022-09-07

**Authors:** Qing Liu, Yongchun Yang, Qingmin Meng, Shan Man, Yidan Wang

**Affiliations:** 1College of Earth and Environmental Sciences, Lanzhou University, Lanzhou 730000, China; 2School of Geography and Planning, Sun Yat-sen University, Guangzhou 510275, China; 3Department of Geosciences, Mississippi State University, Starkville, MS 39762, USA

**Keywords:** urban globalization path, urban localization path, multiple cooperation mechanism, government intervention, resources orientation, Dunhuang

## Abstract

Currently, urbanization driven by global capital flows entails a main trend in many large cites in China, while global capital investment in small inland cities especially in western China is extremely scarce, where their globalization characters the powerful nationalization power and market activation. Dunhuang, a small inland city in western China, has transformed successfully from an agricultural county to an international tourist city, a platform for worldwide cultural communication, and a node city in the Belt and Road region because of its unique and brilliant resources: Mogao Grottoes and Dunhuangology. Therefore, this paper develops a conceptual framework of the multiple cooperative mechanisms and globalization path (MCMGP) of Dunhuang, elaborating the process of industrial transformation, urban globalization, and multiple cooperative mechanisms between government and market actors based on interviewing records and statistics. Findings show that the MCMGP features government-led intervention, resource orientation, and centralization that embodies the driver of state-owned enterprises (SOEs). Also, the MCM in Dunhuang’s globalization contains the mechanism of enrolment, mobilization and action, governance and global marketing, distributed in the two phases. Equally important, in response to the Belt and Road Initiative (BRI) and Silk Road (Dunhuang) International Cultural Expo (SRDICE) from the state, the city government has significantly reinvested and refined cultural tourism via governance mechanisms, carving out a key node city in the Silk Road and elevating an international tourist city. Environmentally, Dunhuang’s tourism internationalization enhances the process of the development of a sustainable shared mobility industry. Furthermore, its tourism development and social–ecology system maintain the synergistic relationships which international tourism promotes such as urban ecosystem and public welfare and in turn, social–ecological enhancement serve Dunhuang’s international tourism well. Practical implications of how Dunhuang’s experience may have lessons for others are discussed in China’s peculiar socialist market economy discourse.

## 1. Introduction

Since the 1970s, global capital flows, foreign direct investment of transnational corporations (TNCs), and the changes and shifts of global production networks have largely accelerated urban globalization development [1]. New research focuses on economic, social, political, and cultural dynamics of urban globalization [2,3]. In general, current research centers on the classic global city studies, which highlights that global cities are the outcome of the new international division of labors, and economic globalization is employed as a core driver for urban globalization by TNCs, advanced producer services (APS), and manufacturing [4,5,6,7,8]. Since 2000, Globalization and World Cities (GaWC) has conducted a lot of research on world city networks [9,10,11,12]; later on, in light of the intersection of the global value chain (GVC) and world city network, some scholars have studied the impact of manufacturing on third world cities and the effects of emerging industrial cities on shaping urban globalization [12,13,14,15]. The developed issue of creating open business innovations for urban globalization [12,13,14,15]. Emphasizing economic, dynamic, and urban spatial reconstruction, in turn, researchers have also begun to focus on the social turn and transnational social space. Global migrants are a key driver to promote urban globalization from the bottom up [16,17], and thus, global cities are possibly centers of international migration [18]. Moreover, ethnic economic zones, settlements, transnational gentrification, and immigration are symbols of social-spatial differentiation of world cities [19,20,21].

Since the 1980s, in-depth urban globalization has been inseparable from the broad support of governments. On the one hand, different types of multinational institutions with complex and diverse global political geographies have been triggered [2,10,22,23,24]. On the other hand, city governments are also a key driver of urban globalization. For example, the development strategy formulated by the city government is a key factor for Shanghai and other first-tier cities in China, when they enter into the ranks of world cities and rapidly improve their status [25]. Similarly, the New South Wales government shaped Sydney’s globalization strategy and promoted its globalization process [26]. Also, the internationalization of national trade, tourism, cultural industries, or other service industries are another important dynamics. The existing literature shows that international cultural exchanges, mega-events, and national flagship projects are paramount ways to promote urban globalization and diplomacy, and they also shape cities’ global cultural identities and reputations [17,20,27,28,29,30,31]. Due to the prolonged slowdown of traditional manufacturing, it is noted that local authorities have made tourism an important strategy of their integration into globalization [32,33]. Existing research in the field mainly focuses on the localized context of tourism, considering tourism as a coupling nexus of globalization and localization [32,34,35]. Noteworthily, there exists a closer relationship between small-town tourism and urban globalization, i.e., Fløysand and Jakobsen elaborate that localities launch commercial marketing of Sogdal Football Club which attracts foreign tourists and promotes globalized tourism in Norway [36]. Likewise, the Salta Wine Region has constructed globalized landscapes, highlighting its high-quality viticulture and global marketing [37].

After revisiting previous research [3,6,7,8,9,10,11,12,17], we find that whether it is a classic world city path as a major economic impetus, or another social, political, and cultural dynamic, the existing literature almost emphasizes that global capital, flow, logistics, and transnational factors (i.e., APS, manufacturing, and global trade) are the original drivers of urban globalization. Unlike the urban globalization path (UGP) of western countries or eastern coastal China within above such contexts, UGP of China’s inland cites has gradually formed the hybrid drivers of “government + SOE + market” under China’s socialist market economic institution, whose path is more dependent on national policies and government initiatives, coupled with necessary resource endowments as the original driver. However, little research has paid attention to such leading roles of resources’ innate triggering factor and government-led multiple synergy [38,39,40,41], which significantly varies from the above UGP which most existing literature focuses on. This study unveils why and how the government nationalized market operation and initial engine of resources are employed and critically expounds its novelty of UGP in China’s inland cities discourse. In doing so, it makes theoretical contributions and refill the gap to the literature dominantly highlighting manufacturing and APS oriented UGP.

Dunhuang, a small inland city in western China has a UGP which could vividly shed light on the above argument. After over 40-years of urban globalization, Dunhuang has transformed from a traditional agricultural county to an international tourist city, a platform for world cultural exchanges, by which it successfully integrates into the globalization process. On the one hand, the unique resource that is Mogao Grottoes is a worldwide Buddhist art storehouse and Dunhuangology is a world cultural heritage tourist destination. On the other hand, it is actively in line with the national BRI and acted as the permanent venue of SRDICE, bridging countries involved in the BRI for cultural cooperation and communication. The local authorities have seized these opportunities and marketed Dunhuang as an international hub city on the BRI, which has profoundly triggered Dunhuang’s modernization and globalization. As a government-led organization path with a cultural engine and unique resource endowment, Dunhuang’s UGP could provide novel insight not only in China but also other countries. Specially, Dunhuang’s local experience could fertilize other cities in two ways: first, it heuristically notes how to internationalize local unique tourism resources, and secondly, shapes the city as an international tourism destination. Eventually, the city’s physical built environment, social-ecological system, public well-being, sustainable environmental awareness and action, service facilities and landscape, and greenery designs appear globalizing turn. More essentially, Dunhuang’s experience is the epitome of fully utilizing nationalization power and market mechanism. Therefore, we would mainly argue: In the context of marginal location and insufficient FDI, how could an initially enclosed market and economy break such path lock-ins and integrate into globalization in a small inland city in China? What kinds of cooperation mechanisms between government and market-related actors should be established? As well as to what extent tourism and social-ecology systems mutually impact and synergistically develop? Concisely, the study further validates and enriches the issue that the locals escalate urban transformation, social-ecology improvement, and globalization change via distinctive resources, and focuses more on the hybrid triggers of “government + SOE + market” under China’s special socialist market economic institution. Below, employing Dunhuang as an empirical study, we will attempt to build a conceptual framework of inland cities’ UGP in China. Then we seize it to expound the process and characteristics of Dunhuang’s globalization, as well as MCM under government intervention. Lastly, we concisely summarize the conclusion and discussions.

## 2. A Conceptual Framework of Urban Globalization Path of Western Cities in China

### 2.1. The Urban Globalization of Western Inland Cities

Since the reform and opening-up policy in 1978, China’s transitional institutions and development levels have formed a zonal gradient strategy from the eastern coast to the northwest inland [20]. A significant large regional gap exists and enlarges in the level of opening to the world and economic development [20,38]. Furthermore, due to factors such as isolated and marginalized locations, underdeveloped and single economy structures with certain national policy restrictions, most small and medium-sized inland cities lack globalized drivers [39]. Through years of research on national gradual institutional changes and western inland cities’ transformations and globalizations, we have found that inland cities have turned to the following three actions since 2000.

There are three representative UGPs of western inland cities. (1) The specialty product trade and industrial internationalization. Natural resource-based cities in western China possess affluent mineral resources, rare metals and special agricultural and sideline products, developing, selling and expanding distinctive and resource-based products and gradually joining the global market via product international trade [40], such as the production, processing, and export of green agricultural products in the Hexi corridor [41]. Also, urban globalization is further undergirded by the industrial internationalization. To be a remarkable representative, SOEs in these cities are embedded in the low-end segment of the TNCs’ GVC, i.e., Jinchang in Gansu Province promotes urban globalization through the overseas investment of Jinchuan Company, a local SOE. (2) The UGP of inbound tourism and international cultural exchange. Many inland tourist cities in western China tend to construct international tourist cities and cultural exchange platforms, where the locals tap into tourism and leisure, unique native culture, and international mega-events to promote urban international image. Some typical cases are Dunhuang mode, Yinchuan mode and, Xining mode in the western inland cities. Yinchuan mode involves an international event: China Arab States Expo, where cultural exchange between China and Arab states via international trade and tourism is booming [31]. Xining mode refers to a major international sports event “Cycling Tour of Qinghai Lake”, coupled with the export of specialty agricultural products, which fosters Xining’s international trade and tourism. (3) The UGP of international trade hubs or entrepot nodes. In some western cities, key transnational transportation hubs situated in the Belt and Road region or the Silk Road Economic Belt exist. The UGP is facilitated by outward trade in their free trade zones, international logistics parks, airports, and railway ports, and is an example of Kashi, an inland city in Xinjiang. More importantly, the similarities of the above UGP in western China may support the relevance of urban-globalized transformations and local social ecology, where environmental and public responsiveness widely exists and intertwined with the local development direction, the social-ecological effects of international tourism in particular (Figure 1).

### 2.2. Multiple Cooperation Mechanism in the Globalization of China’s Inland Cities

With the worldwide prevalence of neoliberalism since the mid-1980s, urban growth and globalization are increasingly dependent on various international capital forces and stakeholders [42]. Especially in western countries, the majority of urban governments increasingly rely on free market mechanisms and attract foreign investment to achieve urban globalization [16,21]. In addition to being generally affected by marketization and globalization since the reform and opening up in 1978, most cities in central and eastern China turn to global capital-driven and expert-oriented UGPs like western cities, whereas the foundation of TNCs, FDI, APS and globalized manufacturing in inland cities of western China is substantially weak [41]. Therefore, they are highly dependent on regional policy supply and government macro-control strategy in the adventure of breaking lock-ins of scarce foreign capitals. Essentially, the UGP of these inland cities is characterized by government-led and intervention, innate trigger of resources (or resource asset), and centralization operated by SOEs’ driver.

The essence of such unique UGP in western cities is manifested in the powerful intervention of government willpower or other public powers. Firstly, owing to the extreme shortage of foreign capitals and investment, the development of inland cities relies on the central governmental policy preference and innate resource endowment. Previously, in the support of “the Third-Front Movement” policy and “the development of the western region in China”, industrial systems of resource-based primary product processing industry and heavy industry have been formed [40]. More importantly, pillar industries in western inland cities are dominantly organized and operated by SOEs that are the core carriers, conducting overseas investment, international trade, tourism project development, and global marketing. Their UGP uniquely features natural resources-oriented development and the synergetic interactions of government and SOEs which partly show the centrality of administrative power under the socialist cities of a planned economy. Such distinctive characteristics and governmental willpower have shaped the multiple-cooperation mechanism of UGP in the context of inland cities in China. First, city governments act as the core driver and organizer in the initial stage of a city’s integration into globalization. The government establishes urban interest coalitions through an enrolment initiative and joint action mechanism amidst stakeholders. Subsequently, as cities enter the growth stage of globalization and marketization, the government tends to fully intervene and regulates the operation of the entire city’s outward market through a centralized governance mechanism. Ultimately, in the mature stage of urban globalization, the locals need tap into and refine the city’s cultural stock and reposition their cultural identity for high-quality development. City marketing led by the government has become the main mechanism for shaping the city’s international image (Figure 1). The above conceptual model will be tested and enriched with the evidence of Dunhuang involving its globalization, localization, and multiple cooperation mechanisms.

## 3. Study Area and Data Survey

### 3.1. Study Area

Dunhuang, located on the largest oasis of the Dang and Shule Rivers and surrounded by the Gobi Desert, is a county-level city in the northwest Gansu Province, China. Its city prefecture covers seven towns and two villages, peculiarly, Shazhou town is the seat of the city government and is the political, economic and cultural center of Dunhuang (Figure 2). Before China’s reform and opening-up since 1978, Dunhuang had always been a backward agricultural county reliant on wheat, cotton, and characteristic fruit plants. It was not until 1979 that Dunhuang began to develop tourism as one of the first tourism opening cities in China, gradually achieving a transition from an agricultural economy to a tourism service economy.

Notwithstanding Dunhuang does not possess a developed industrial foundation, it attracts a worldwide preoccupation due to its unique cultural and natural assets. In the sphere of human and historical landscapes, there are 265 cultural heritage sites in Dunhuang, three of which are World Cultural Heritage sites, the Mogao Grottoes, the Yumen Pass, and the site of the Xuanquanzhi Posthouse. Specifically, Mogao Grottoes is the largest- and richest-surviving Buddhist art holy site with mysteriously “Oriental” charm, bearing the most continuous historical Buddhism preservation worldwide. Besides, Yumen Pass and Yangguan Pass are strategic passes on the Silk Road from Dunhuang to the north and south of the ancient Western Regions, respectively. The site of the Xuanquanzhi Posthouse is the seat of the administrative institutions, postal system of Dunhuang during the Han Dynasty (94 BC) in China, where massive unearthed vestiges and remains are priceless materials for archaeologists to study the history of the Han Dynasty. The ancient Dunhuang city, also called a movie and television city, is designed and built by the Japanese imitating the ancient Shazhou City of the Song Dynasty in China, where the Sino-Japanese jointly shot the grand historical film “Dunhuang” in 1987. As for natural landscapes, Mingsha Mountain Crescent Spring is a rare desert spectacle in the world, it is commonsense that deserts and clear springs rarely coexist, thus, such a unique landform combination of mountains and springs has enhanced the irreplaceability of Dunhuang’s tourism resources. Moreover, Yardan National Geopark is the largest aeolian landform group in Asia, which is also called Demon Castle due to its breathtaking desert wind-eroded landscape. It is the setting for many film and television works and an important venue for off-road and auto rallies (Figure 3). In short, there exist culturally rich and naturally blessed resources in Dunhuang. Finally, as for distribution of tourism resources and travel routes within Dunhuang, major routes are divided into eastward and westward routes, respectively. To be specific, the Eastward Route covers urban area tourism and is the core of the entire Dunhuang tourism industry, while the Westward Route is a suburban tourism line.

### 3.2. Data and the Survey Design

The research data mainly include two parts: (1) the field survey data and in-depth interview data in Dunhuang. We conducted semi-structured interviews with respondents through an interview questionnaire, recorded in the form of audio, and on-site conservation notes, and finally we organized the recorded notes as the first-hand material. Specifically, respondents encompass officials and staffs of the city government and its related sectors (i.e., Foreign Affairs Office, Development and Reform Commission, Political Consultative Conference, and Civic Center), Dunhuang Tourism Board, Dunhuang Academy China, SRDICE Bureau, SRDICE’s International Conference and Exhibition Center, Management Committee of Scenic Spots, Dunhuang Cultural Tourism Group (SOE, CTG), international tourism agency, and local residents. (2) Statistical data and planning documents of Dunhuang as second-hand data, include Dunhuang Statistical Yearbook (2010~2019), Dunhuang National Economic and Social Development Statistical Bulletin, Dunhuang Academy China Yearbook (2005~2017), Dunhuang 13th Five-Year Plan, historical and cultural city protection planning documents.

Primary data were acquired from 19-day incessant field trips in Dunhuang from 28 September 2020 to 15 October 2020, coupled with follow-up telephone interviews. The content encompassed interview outlines on Dunhuang’s globalizing mechanism and survey questionnaires on residents’ perception of Dunhuang’s globalization. Concretely, the interview outline contains the typical period of Dunhuang’s foreign exchanges, storylines, landmark events, distinctive performances, paths, mechanisms of Dunhuang’s globalization in different periods, and the motivation and experience of tourists visiting Dunhuang. The questionnaires on residents’ globalizing perception of Dunhuang employ 6 dimensions and 21 indicators: globalized cultural identity, city image, cultural exchange, transportation and communication, urban landscape and physical environment, and tourism uniqueness. In term of the structure of respondents, we were acquainted with the main participants in Dunhuang tourism construction and urban globalization, and then eventually determined our six respondent subjects, related government departments, Dunhuang Academy China, CTG (SOE), international travel agencies, local residents, and international tourists. During the process of investigation, two students formed group during the formal survey, one was responsible for the interview with the respondents, the other was responsible for audio recordings and taking notes of key information on site, then the recordings and notes were sorted into electronic texts according to a corresponding questionnaire number. Consequently, 52 written interviews, each taking half an hour to 1 h, and 346 valid questionnaires were obtained. Moreover, the author applied anonymous coding disposal later to protect the privacy of respondents. “####Y*” as coding form appears in the paper, the four #s represent month and day of the interview, Y is the code for respondents’ identity, * indicates respondents’ order number. Respondents’ identity is coded as below: A shows local residents, B represents staff of government and tourism-related departments, C specifies staff of Dunhuang Academy China, D expresses the commercial service operators, E is for tourists, and F indicates tourism developers. For example, “0931A1” means the first local resident interviewed on 31st, September.

## 4. Industrial Transformation and Tourism-Driven Globalization in Dunhuang

### 4.1. Tourism Ascendancy, Industrialization and Transformation

To date, Dunhuang’s industrial structure has emerged with significant transformation since Dunhuang developed tourism. As shown in Figure 4a, the three industrial sectors’ GDP and their proportions dramatically changed from 1970 to 2020. Initially the primary industry accounted for the highest proportion of GDP from 1979 to 1997. However, the figure has been decreasing, while those of GDP in the secondary and tertiary industry have been creeping up. Subsequently, there exists a remarkable turning point in 1998, when GDP of tertiary industry exceeds that of the primary for the first time. Furthermore, the data of tertiary industry soars from 1998 to 2007, whereas that of primary industry appears to plunge. Eventually, the tertiary industry became overwhelmingly dominant in the economic development during 2008~2019. Next, Figure 4b depicts the distribution of the three industrial sectors’ added value and the proportion of industrially active persons during 2010~2019. Detailed information suggests that: primary industry’s added value presents the lowest proportion and the smallest span, which is located in the 12~19 interval, while the figure of tertiary industry reaches the highest proportion and largest span, ranging from 49 to 68. As for the proportion of industrially active persons, the average distribution proportion in the three industries is 38.48:15.01:46.51. The employment structure of Dunhuang mainly consists of employees of agriculture and tourism services whereas that of manufacturing employees levels off at a low level.

Inbound tourists played a leading role to reinvigorate the tourism industry in Dunhuang. First, inbound tourists of Dunhuang are mainly from foreign countries such as Chinese Hong Kong, Macau, and Taiwan (Figure 5a). Briefly, foreign tourist arrivals have seen a marked ascendancy, followed by Taiwanese tourists, and Hong Kong-Macau’s tourists. To further demonstrate, the average proportions of the three types of tourists to the total inbound tourists are 63.53%, 13.68%, and 22.79%, respectively. As for the dynamic trend, foreign tourist arrivals reach the highest peak in 2011~2013 and the second peak in 2017~2019. Hong Kong-Macau’s tourists, by contrast, stabilize at an overall low level, and Taiwanese tourists tend to continuously augment. Third, as for foreign tourist source countries and market structure (Figure 5b), Japan is the largest source of foreign tourists, accounting for the absolute dominance in Dunhuang’s foreign crowds, followed by both South Korea and the United States. Furthermore, Australia, Singapore, Malaysia, the United Kingdom, and France are also the main source countries. Dunhuang’s foreign tourists mainly focus on East Asia, Southeast Asia, and Western Europe.

### 4.2. Tourism-Driven Urban Globalization in Dunhuang

The urban globalization of Dunhuang originates from tourism opening to the world and international communications of Dunhuang culture. Having witnessed its tourism development over the past 40 years, Dunhuang has developed and formed a sophisticated tourism market, which has been depicted clearly by four phases in its process of tourism development and urban globalization (Figure 6).

Stage I: The booming phase of inbound tourists and tourism industry growth (1979~2000). Since China founded in 1949, Dunhuang was a traditional agricultural county. Later, Dunhuang was designated as one of the first tourism opening cities to the world. By the state. In fact, as early as the 1900s, the discovery of the Library Cave of Mogao Grottoes arose as a severe hit worldwide, after that, plenty of cultural relics, Dunhuang Manuscripts, and Documents of Mogao Grottoes were scattered to many western countries around the world, and the consequent preoccupation with Dunhuang surged due to Mogao Grottoes. Therefore, before the Dunhuang tourism opening to the world, Mogao Grottoes and Dunhuangology had been vastly expanded beyond their shores. Since inbound tourists were able to enter Dunhuang in 1979, Dunhuang has been flooded with spontaneous large-scale worldwide tourists who are eager to witness the rare Buddhist art treasure trove, and overriding arrivals emanated from Japan. As a scholar of Dunhuang Academy China (1010C2, Director of the research department) said: “what attracts most the surge of inbound tourists is Mogao Grottoes, due to its own unique brilliant historical continuity alongside over 1600 years of elaboration history, the massiveness of this culture is at the top in the world. Especially, Japanese have a special affection and attachment for Mogao Grottoes. Dunhuang is consider as their cultural homeland, and coming to Dunhuang is essential for their root-seeking tourism.” A tourist (1002E3) said: “grand statues, elegant apsaras, amazing murals, exquisitely arranged caisson ceilings of Mogao Grottoes, holy world of Buddhism, secular life, the hardships of ancient silk businessmen and the luxury of the Western Regions palaces a thousand years ago were all vividly visible in Mogao Grottoes, which shocked me a lot.” In addition, the UK’s BBC, Japan’s NHK TV, and China’s CCTV jointly filmed the documentary “Silk Road”, Dunhuang was one of the vital nodes, and the documentary contributed to a global sensation. The UNESCO listed Mogao Grottoes on the World Cultural Heritage List in 1987. The film “Dunhuang” co-produced by China and Japan premiered in 1988. These international events have greatly established and enhanced a global presence of Dunhuang.

Localized reforms of institutions, industries, and related services are conducted the step-by-step urban globalization For example, to serve foreign guests well, the city government established the Foreign Affairs Office in 1979, with its branched China International Travel Agency, and Dunhuang Hotel, the three departments are jointly in charge of the reception and hospitality of foreign guests. An office director of Dunhuang Tourism Board (0928B2) said: “the tourism industry in Dunhuang was referred more as a diplomatic affair at that time, foreign tourists dominated the whole tourists, so the Dunhuang Tourism Board also evolved from the past Foreign Affairs Office.” As a local resident (1005A2) said: “with Dunhuang’s tourism opening to the world in 1980s, ever-increasing foreigners came here because of Mogao Grottoes and its Library Cave. We were lingering curious about foreigners. It was not until the mid 1980s that we often met foreigners, and we became gradually accustomed to them”.

Stage II: The policy-driven development of tourism destination (2001~2008). Dunhuang experienced a peak period of foreign tourists in the 1980s~1990s. But Dunhuang’s infrastructure failed to meet tourists’ surge in demands at that time. Also, Suoyang City, Yumenguan Pass, and the Site of the Xuanquan Posthouse in Dunhuang were successfully nominated for the World Heritage Sites in 2006, plus Mogao Grottoes and the Ruins of Great Wall. Now there are five World Cultural Heritage Sites in Dunhuang to date. Subsequently, the city government proceeded with forming the “Tourism Nurturing the City” strategy. Under the strategy of tourism-based urban development, the city government gradually envisioned a blueprint strategy of “Tourism Nurturing City”, and then it laid stress on Dunhuang’s tourism, which became the invigorate pillar industry.

Since 2001, the government has released various types of tourism development plans as a buffer to sustain Dunhuang’s tourism, vigorously building tourism-supporting facilities, strengthening tourism management, and encouraging local residents to participate in the tourism market. (1) Transportation system: The government upgrades and updates the transportation systems, accommodation systems, and the related tourism industry management. The government has expanded the capacity of Dunhuang Airport three times, Dunhuang International Airport was formally opened with international airlines in 2007; the Dunhuang Railway and other roads from the urban area to various tourist attractions were large-built and opened to traffic. (2) Accommodation: in addition to attracting foreign investment to build star hotels in Dunhuang, the city government encouraged local residents to build pubs or inns. (3) Tourism industry management: the city government established the Dunhuang Tourism Management Committee that was in charge of the management, supervision and evaluation of Dunhuang’s tourism market. (4) The makeover of cityscape and environment: the mayor S made abundant reformations during his tenure, such as the renovation of the cityscape and environment, the makeover of the Danghe River Landscape Line, the exploitation of tourism projects, and etc. A government official (Director of development and reform commission, 0929B1) said: “Dunhuang’s urban environment was deadly poor previously. Mayor S considered that such environment was incompatible with an tourist city image. Dunhuang was deemed a world-famous tourist resort, and antiseptic city environment and civilized behavior of the populace should be developed as a mirror of its tourism reputation. So he carried out voluntary cleaning exercise of urban garbage, the sewage treatment and Danghe river landscape construction, and appealed to the public improving tourism service awareness”. Another local resident (1006A5) said: “There existed a key time node in 2005~2006, cityscape and environment appeared dramatic amelioration. Dunhuang was not always a tourism city that only relies on Mogao Grottoes to attract tourists since then. Mayor S said that our livelihood were directly linked to Dunhuang tourism, only when we did well in urban sanitation and landscape can Dunhuang attract ever-increasing tourist arrivals. Hence, everyone was enthusiasm and took care of the cityscape as a habit”.

Stage III: Centralized organization and operation of the tourism market (2009~2015). The city government attempted to reorganize and optimize Dunhuang’s tourism market, the core of which is centralized governance of city-owned tourism resources. Detailed action was launched as follow: (1) The city government has devised various development and protection plans, in addition to master plans of the Silk Road Economic Belt, aiming to strengthen the conservation of world cultural heritages and the construction of urban landscape. (2) The germination stage of international tourism festivals in Dunhuang City government has six-times hosted the Dunhuang Apsaras International Cultural Tourism Festivals since 2000. In 2013, the United Nations World Tourism Organization hosted the Silk Road International Tourism Festival and International Conference in Dunhuang; the city government held the Dunhuang International Music Festival, International Cultural Relics Expo, and International Urban Sculpture Exhibition in 2014~2015. (3) The construction stage of Dunhuang’s Smart Tourism. Dunhuang was listed as a smart city pilot by the State Council in 2013, which largely prompted the construction of smart tourist attractions and the digital protection mode in Dunhuang. (4) The stage of centralized operation and governance of city-owned tourism resources. The city government successively established operational companies for each tourist attraction with smart management systems since 2014. Recently, the government established a SOE, Dunhuang Cultural Tourism Group (CTG), a core carrier about centralized operation of city government. A deputy director (Administration Department of CTG, 0931F1) said: “The city government integrated other tourism resources except Mogao Grottoes, and seized the SOE mode to collectively govern Dunhuang tourism. In practice, the mode has markedly altered previous Dunhuang tourism market dominated by folk spontaneous behaviors, since then the tourism market has been stepping into a standardized and sophisticated stage. Moreover, the reason why the government set up CTG to start centralized pilot is that the government aim to establish a market-oriented tourism system”.

Stage IV: International mega-event marketing and government-driven cultural exchange (2016~present). In 2013, Chairman Xi Jinping of PRC laid out “BRI”, aiming to promote international cooperation by joint actions between China and the “Belt-and-Road” regions. In response to this national strategy, the city government began to carve out a specialized niche as an international tourist city and a hub of on the Silk Road by the world-renowned Mogao Grottoes and Dunhuangology. With the approval of the state, SRDICE has been held in Dunhuang annually since 2016, and Dunhuang became the permanent conference site. In 2016, six foreign state leaders and former state leaders attended the Open Ceremony of SRDICE, and 95 delegations from 85 countries, five international organizations or NGOs, 66 foreign institutions, 434 foreign guests and 1700 Chinese guests participated. Finally, the official “Dunhuang Declaration”, a programmatic file with high-level intergovernmental cultural dialogue, was jointly issued. In this period, the government-led preparatory work markedly boosted the globalization of Dunhuang: (1) The city government expanded Dunhuang Airport and multiple international air routes were added. (2) The renovation and upgrading of Dunhuang international or star-rated hotels were accelerated, and more developers were attracted to invest high-end hotels. There are 243 newly built hotels in Dunhuang, which is much higher than the total number of the previous stages. (3) More significant international exhibition venues and related facilities of SRDICE were conducted. Specifically, the city government built the Silk Road International Convention and Exhibition Center after the regional government released the planning scheme in 2013.

### 4.3. Tourism Impacts on Social-Ecological Effects in Duhuang

Benefits in the urban social-ecology system induced by tourism are assessed as illustrated in Figure 7. Dunhuang’s tourism development and social-ecology system maintain the synergistic relationships, whose intersection is a two-way interplay process. For one front, international tourism impacts urban ecosystem and public welfare. Environmentally, tourism development has elevated the following four aspects: the renovation and amelioration of human inhabitant environment (i.e., cityscape, built environment, greenery, amenity), the sustainable development of natural environment and resource capacity, the preservation planning of cultural heritages (Mogao Grotto in particular, like cave refurbishment, the monitor of mural diseases, and pests and microenvironment in caves, desertification governance), and the optimization of industrial structure (reduction of heavily pollution-creating industries and the augmentation of the eco-friendly service sector). For the indigenous populace, international tourism has enhanced their public well-being portfolio diversification as a major sustainability generator in three pathways, those being: the livelihood sustainability of destination communities (pro-poor tourism, increased local employment and income induced by tourism, and the retention of labor), the sustainability of healthy lifestyle, and life quality and food security coupled with sustainability of environmental consciousness and joint action. Furthermore, the above social-ecological enhancements as one agent serve Dunhuang’s international tourism well. Dunhuang’s tourism is inextricably interwoven with urban environmental health and public wellbeing (Figure 7).

## 5. Multiple Cooperation Mechanism (MCM) of Urban Globalization

### 5.1. Related Actors in the Multiple Cooperation Mechanism

The related actors of the city government of Dunhuang are the Dunhuang Academy China, tourists, local populace, international travel agencies, and tourism developers (CTG). The details of related actors are: (1) The city government: Previously, Dunhuang was predominantly equipped with an agricultural economy but the harvest has become stagnant. In response to the state’s appeal of tourism opening to the world, the city government establishes the Dunhuang’s transformation route with tourism as its leading industry. Specifically, in order to bolster struggling urban economies and melt Dunhuang into a globalization process, the city government utilizes Mogao Grottoes and other tourism resources to expand its international tourism market, formulates tourism strategies and plans, renovates urban environment, builds infrastructure, and implements national policies. (2) The Dunhuang Academy China: In reaction to state government’s conservation strategy for Mogao Grottoes, The Dunhuang Academy China was established and takes responsibility of protecting, studying, promoting, and managing the World Cultural Heritage Mogao Grottoes, with its affiliated West Thousand Buddha Grottoes and Yulin Grottoes. The techniques of repairing the ravages, permanent preservation and cultural transmission are the primary preoccupations. (3) International travel agencies: limited by its limited urban size, local travel agencies are clustered, and rather isolated conditions of transportation and communication for themselves and foreign tourists. (4) The CTG: its main task is to pool city-owned resources for centralized management and operation, and advance smart tourism and cultural tourism projects. (5) Local residents: There exists recurrent droughts, sandstorms, and water shortage in the wake of substantially low agricultural land utilization. Peculiarly, local cottons threatened by long-staple cotton in Xinjiang province of China have encountered unsalable dilemma since 1990s, which largely affected residents’ livelihood due to a flagging agriculture sector. Fortunately, they have recognized that they could improve their income and expand employment opportunities by the participation in the promising tourism industry. (6) Tourists: as for global scholars and academic groups with strong interests in Dunhuangology, Buddhist culture, grotto art, architecture, music, etc., there are few destinations that have been constantly elaborated for 1600 years like Mogao Grottoes.

### 5.2. Formation of Multiple Cooperation Network for World Cultural Heritage Site Construction

The globalizing process of Dunhuang indicates that its urban transformation and globalization can be divided into two phases, “Construction and Operation”. (1) The World cultural heritage destination construction (1979~2008): Dunhuang has been converted into a typical World Heritage Site, and a worldwide tourist city with escalating infrastructure. (2) The centralized operation and global city marketing (2009~present): drawing on BRI and the international platform of SRDICE, the city government spreads Dunhuang culture worldwide and carves out an international cultural tourist city. More details are as follows (Figure 8).

Four types of enrolment can be summarized in the formation of MCM. (1) Administrative enrollment: The incentives for the world cultural heritage destination construction of Dunhuang were initiated by the central government, which enrolled the local government of Dunhuang to implement all aspects of tourism deployment. Additionally, Dunhuang Academy China preserves and nurtures cultural heritages of Mogao Grottoes since it was founded in 1944. Ever since Chairman Deng Xiaoping visited Dunhuang Academy China and Mogao Grottoes in 1981 and the successive national leaders all came here for inspections. Chairman Deng Xiaoping said: “The conservation of Mogao Grottoes is a pressing matter, Dunhuang’s cultural relics are world-famous and the priceless treasure, which must be protected by all means”. Chairman Xi Jinping said: “We must well preserve our national quintessence and make Mogao Grottoes an excellent model for heritage conservation (cited by the People’s Daily Overseas Edition)”. Under the instructions of the successive national leaders and their conservation policy, Dunhuang’s heritage conservation has been translated into tangible results through unremitting efforts of the Dunhuang Academy staff led by successive deans Chang Shuhong, Duan Wenjie and Fan Jinshi. (2) Environmental enrollment: since 1982, in order to build a world cultural heritage destination equipped with pleasing tourism infrastructure and physical environment, city government has enrolled every town government to add “soft touch” and “refinement” to urban landscape. (3) Image marketing: to enhance Dunhuang’s international presence, the city government enrolled related actors to actively participate in the tourism industry and to market Dunhuang tourism by online and actual means. (4) Unique resource enrollment. Dunhuang’s well-endowed world cultural heritages and miraculous natural landscapes have attracted the BBC, NHK, and CCTV to film documentaries, movies, and TV series. Massive domestic and abroad scholars and teams interested in Dunhuangology have studied and written scholarly monographs, papers, novels and poems.

Three types of mobilization mechanisms were conducted. (1) Incentives and financial supports: The city government issued relevant policies to encourage other actors to supply tourism services. First, the city government encouraged residents in urban areas and near scenic spots to build hotels, inns, or pubs (Figure 9). Aside from that, town governments provided financial support to farmers who planted grapes, apricots, and jujubes and raised camels, excellent fruit growers and camel keepers were honored and rewarded by local governments. A camel keeper in Yueyaquan Village (1005A7) said: “In the 1980s, Yueyaquan villagers predominantly planted fruits, but had a meager income. With tourism opening to world and massive foreign tourists flooding to Dunhuang, some villagers utilized their household camels to run the business. Tourists were keen on riding camels across the desert, or taking photos about their camel team riding with our help, which has rendered a considerable income source. Thus, an emerging camel-rising industry was booming in this area. Finally, after riding camels, tourists were prone to buy Dunhuang souvenirs around scenic spots, like carpets, silk scarves, scriptures and silks, wood carvings of Buddha statues, and flying murals, these all prompted the formation of a complete industrial chain”. Second, Financial supports to conserve Mogao Grottoes: The central government and provincial government have continuousl administrative financial allocations. Other than this, Mr Ikuo Hirayama, president of Tokyo University of the Arts, established the Ikuo Hirayama Academic Foundation and China Dunhuang Grottoes Conservation Research Foundation, and Ms Niemi Gates established the American Dunhuang Foundation to support the Dunhuang Academy China for protective research on Mogao Grottoes. Third, the involvement of international travel agencies: since 2006, the city government released loan operation licences and business registration for international travel agencies, they have been flourishing (Figure 9). (2) Cityscape makeover: From 1979 to 1985, the city government launched the first round of urban greening construction. In addition, Mayor S set up the City Environment and Landscape Management Bureau, and it was in charge of the greening projects covering residential areas, streets, and attraction sites.

(3) International communication. Firstly, the city government takes communication of Dunhuang tourism seriously, through a combination of physical media, promotion meetings in various cities and international sister cities, and virtual media such as marketing films, global social media (Facebook, Twitter, Tik Tok, and Instagram), and press coverage in addition to government websites, which have greatly enhanced and expanded the city marketing power. Most notably, Dunhuang Academy China was still the dominant actor transmitting Dunhuang culture to the world. Detailed approaches are summarized: (a) International cooperation projects (cave conservation, digitization, academic training, and etc.): Dunhuang Academy China conducts joint studies in Japan, as well as conservation postgraduate exchange projects and technical collaborations with University of Illinois, University of Oxford, University of London, University of Nottingham, Osaka University. As for cave digitization, Japanese government funded the construction of the Dunhuang Caves Conservation and Research Exhibition Centre in 1994; Dunhuang Academy formally began digital study of Mogao Grottoes s in 1998 in collaboration with the Mellon Foundation of the USA, followed by the International Dunhuang Project (IDP) with the British Library about international standard digitization work of Dunhuang scriptures and cultural relics. (b) International exhibitions: Since the mid-1940s, Dunhuang Academy China on its own or jointly with international institutions has been exhibiting globally in 37 cities worldwide, involving murals, cultural relics, scripts, and cultural products. (c) International academic exchange and hospitality: It mainly includes overseas visits and international conferences participated by researchers of Dunhuang Academy, international conferences and forums and foreign scholars’ lectures at Dunhuang Academy, and the hospitality of foreign official guests, national delegations, international organizations, academic teams, and scholars. (d) Digital Dunhuang and International website marketing: For decades, Dunhuang Academy collaborated with many domestic and abroad research institutions to preserve the precious materials of Dunhuang Caves using computer digital technology. Moreover, Dunhuang Academy set up its own global website and digital Dunhuang website, making a significant contribution to the promotion of Dunhuang culture.

### 5.3. City Government-Led Centralized Operation and Global Marketing Phase: The Pursuit of International Cultural Identity

In this phrase appears the changeable intentions of key actors and governance mechanisms (Figure 10). After erstwhile efforts, Dunhuang formed a preliminary but less-orderly tourism market guided by the city government and multi-participants spontaneously, which need to be managed and refined for a more sophisticated tourism market. Besides, in wake of BRI and SRDICE, urban prospect should also focus on Dunhuang’s globalized reposition. In-depth Dunhuang culture mining and enhancement, city marketing were put on the agenda.

#### 5.3.1. Governance Mechanism

The establishment of Dunhuang CTG is the milestone of governance mechanism reform. Since 2013, the city government integrated ICT infrastructure, urban management, transportation planning, and other tourism assets into the Dunhuang smart tourism system based on Smart Tourism Co., Ltd. as a pilot item. A staff of the human resources Department of CTG said (0931F2): “Recently, tourists almost search a public WiFi signal called “zhihuilvyou” during their travels in Dunhuang thanks to government’s vigorous efforts to project smart cities construction, which greatly advances the online experience of tourists”. On this premise, the city government established a new SOE and CTG, and started a market-oriented operation mode of tourism resources. Firstly, the mode has brought reforms to Dunhuang’s international travel agencies. Erstwhile enterprises like Dunhuang CITS have gradually faded, whereas emerging “Internet +” international travel agencies represented by Chenguang Smart Tourism have vastly appeared in Dunhuang. Secondly, CTG has a stringent and meticulous supervision on the entire tourism market, i.e., it distributes camel numbers to the camel keepers in Yueyaquan village and Mingshan village, and establishes electronic archives of camels according to the number; it renovates and rebuilds retail and stall outlets of the Shazhou Night Market, and then gives shopkeepers respective numbers for smart regulation; it projects plenty of scenic spot activities, such as a car rally in Yandan, sand skiing festival, desert camping, Gobi hiking, motorcycle surfing, etc.

Cultural tourism governance is introduced in wake of SRDICE. Currently, the site of SRDICE is permanently settled in Dunhuang, which could offer a superior international communication springboard. In fact, it is emblematic of new departure for Dunhuang’s cultural tourism where distinctive cityscape and emerging industries vastly emerge. (a) Refashioning urban landscape: The city government always explores how to commercialize Dunhuang culture and repackage cultural landscapes by tapping new cultural visual elements and narratives into urban spaces. A staff of the Development and Reform Commission in the city government (1009B5) said: “since first SRDICE, the city government attempts to infuse a sense of historicity in the cityscape and built environment drawing upon Dunhuang culture. In a similar vein, urban landscape like floor tiles, street lights, information plaques and signages, guardrails, seats, are embedded in cultural elements of Mogao Grottoes including flying apsaras, rebound pipa, caisson ceiling, nine-color deer, lotusbricks, and the city brims over with cultural ambience”. (b) Emerging cultural industries. First, Dunhuang’s cultural and innovation industry begun to flourish since SRDICE. The CTG holds Dunhuang international design week and collects design schemes incorporating Dunhuang elements from global designers during SRDICE, then it purchases the IP rights and puts them into further production. Afterwards, the artefacts will have been sold at the physical and online stores. Also, the Center of Silk Arts Fan Yanyan are set in Dunhuang museum. The brands are collected by lots of international senior politicians and celebrities, which bridges culture communication worldwide (Figure 11). Second, SRDICE has incubated the “conference and exhibition industry” and arisen a huge “SRDICE effect” annually. A director of the SRDICE Bureau (1012B11) puts it: “we invite senior national leaders, entrepreneurs, scholars, organizations and medias worldwide, and they participate in international forums, cultural relics fairs, sign cooperation and exchange projects, introduce regional investment and trade, and at last, which greatly enhances the global presence of Dunhuang. Also, such a large-scale mobility of people flow must bring a series of consumption demand, like accommodation, catering, transportation, shopping, as well as venue rental for holding activities, which directly or indirectly stimulates the growth of entire urban economic”.

#### 5.3.2. Functional Key Actors

In the construction phase of the world cultural heritage destination, the city government and Dunhuang Academy China were key actors. The government acted as a guide for the local tourism market and provided various types of offerings for other actors. Dunhuang Academy China carried out the restoration, conservation research, and cultural promotion of Dunhuang Grottoes. Then, in the second phase, the city government repositioned itself and then formed Dunhuang CTG, where city government has completed centralized governance by resource reorganization, culture enhancement and infrastructure support. Afterwards, with the emergence of BRI and SRDICE, the city government officially started globally marketing and repackaging local culture. A staff of the SRDICE Bureau (0929B6, Office Director) stated, “Dunhuang’s tourism has started since the 1980s, nonetheless, its tourism always remains at the level of spontaneous non-governmental communications initially. It has only been since the SRDICE that single model was altered. Specifically, the concept of human Dunhuang and world Dunhuang is formally laid out in the form of global intergovernmental dialogues. Later, an official, multi-participant international cultural communication has been formed, which further magnifies global influence of Dunhuang”.

#### 5.3.3. The Functions of Different-Level Government and the Characteristics of Dunhuang Tour-Ism-Oriented State-Owned Enterprise

Generally, the national government is responsible for macro-control strategic deployment, regional government forms a connecting link between centrals and locals, acting as a coordinator/mediator; local government is the direct operator and executor of tourism activation. (1) The will of national government focuses on three aspects locating the main direction of Dunhuang’s development. Most notably, it prioritized the conservation of Mogao Grottoes, the inheritance and international promotion of Dunhuangology fostering China’s Dunhuangology research into international frontier. Later, it allowed that Mogao Grottoes was accessible to foreign tourists and designed Dunhuang’s trajectory of the international tourism change since China’s reform policy in 1978. Finally, grounded on BRI, it launched the scheme of holding SRDICE annually with Dunhuang acting as the permanent venue, bridging the Belt and Road region. (2) The regional government is more responsive in configuration to the central government’s policy. Initially, it positioned Dunhuang Tourism as the core of the whole regional tourism and provided policy initiatives and software and hardware facilities. Then, it formulated Dunhuang’s all-for-one tourism development plan and historical and cultural city conservation plan. More noteworthily, it is the host unit of SRDICE and initiates SRDICE Bureau in charge of coordinating and operating the international event. (3) In addition to responsive actions to its superior government, local government also carried out extra autonomous activation. To start with, it supplemented alot of tourism-related departments, and cooperated or assisted with various works of the Dunhuang Academy China. Moreover, it energetically advanced the cityscape, physical environment, the residents’ tourism hospitality and etiquette, and etc. More importantly, the local government pooled city-owned resources except the Mogao Grottoes, and initiated CTG, whose duty includes projected global smart tourism, city marketing and social media promotion.

Compared with other SOEs in the traditional China’s discourse, like resource-based SOEs (oil, railway, natural gas, electricity, and etc.) and SOEs of pillar industry for national economy development (electronics, automobiles, pharmaceuticals, telecommunication, aerospace science, and etc.), CTG has more market vitality and flexibility. Likewise, CTG is still transformed from the initial government department, but more innovative business modes and IT technology are applied. In particular, CTG grew out of Dunhuang Smart Tourism Company whose core business consisted in Internet, cloud computing, IoT, big data, and Dunhuang tourism promotion on social media and tourism-related apps (i.e., Facebook, Tik tok, Bilibili, YouTube, Yahoo! Travel, Ctrip, and etc.). Thus, the local government centralizes and activates state-owned resources based on a more innovative and vigorous Internet company, and adopts a novel operating model of SOE that combines online and offline, belonging to a relatively successful local experience of national tourism operation that also inspires some traditional tourist cities or SOEs.

## 6. Conclusions

The main conclusions of the work can be reached: (1) essentially, the UGP of inland cities is characterized by the drivers of more government-led centralization and intervention, innate trigger of resources, and centralization by SOEs. (2) Dunhuang’s UGP evolves from four stages, the development clue is that Dunhuang’s world cultural heritage tourism is initially driven by global visits and international hospitality. Subsequently, the international tourism boom promotes notable urban transformation and social-ecology renewal. Ultimately its cultural stock repackaging feedback afresh urban tourism and marketing. (3) As an environmental reflection during globalization, it is pointed out that tourism is not an isolated activity but it interacts with local social-ecology systems, forming synergistically supportive relations with environmental health and public well-being. Meanwhile, ameliorated social-ecology systems could well serve and foster international tourism. (4) The MCMGP of Dunhuang entails enrollment mechanism, mobilization and action mechanism, governance mechanism, and global city marketing mechanism. These mechanisms distribute in two phases: world cultural heritage destination construction and centralized operation and global marketing. In the former phase, the global presence of Mogao Grottoes and Dunhuang culture attracts numerous inbound tourists. City government and Dunhuang Academy China, as key actors, mobilize related actors bonding common interest via various enrollment methods, and put Dunhuang’s tourism “hardware” in place drawing upon action mechanisms. And (5) in the latter phase, the city government organized governance mechanisms, and established the CTG to project smart tourism and conduct centralized operation by pooling city-owned resources. In response to BRI and SRDICD, city government vigorously nurtured cultural tourism via marketing mechanism, and Dunhuang is marketed as a node city on Silk Road and an international tourism city seizing the marketing tagline “Human Dunhuang, World Dunhuang”. The city government acted as a key actor, which is a typical governance process from decentralization to centralization. Furthermore, Dunhuang tourism has also shifted from non-governmental folk or academic communications to an intergovernmental dialogue form.

Most notably, Dunhuang’s indigenous practice actually exists such as relatedness and enlightenment compared to other practices in the world. (1) International tourism is supportive to urban sustainable development transformation associated with enhancement of residents’ sustainable livelihoods [43]. Nunkoo argued that tourism is an eco-friendly and smokeless industry relative to traditional economic ones [44]. It is also beneficial to facilitate the industrial structure optimization as consequence of replacement of heavy-polluted industry [45]. More importantly, it is useful and practical to guide exploration of the contribution of tourism to residents’ livelihoods via diverse employment and income availability [43]. Overall, Dunhuang’s international tourism and social ecosystem have also formed a virtuous circle of human-land relationship, further have fostered social and environmental justice. (2) Dunhuang’s international tourism promotes urban diversified system, which pertains to a microcosm of urban globalized transition. In Golba’s and Turoń’s productive works, diversity management has widely spread with more and more attention to attach organizations in all industries, Transport-Shipping-Logistics in particular [46,47]. Similarly, Dunhuang’s international tourism has also diffused to all walks of life in this work, mapping multi-faced social capitals into the synergistic diversity ecosystem. Concisely, employment diversification, business diversification, industrial chain diversification, cityscape diversification, culture stock diversification, governance and operation diversification, residential lifestyle diversification, etc. (3) Dunhuang’s smart tourism is one of the most vivid representatives of open innovation business modes. In effect, from the angle of open-shared mobility and sustainable transport system, the issue of creating open business innovations has substantially developed in Turoń’s seminal research [47]. Meanwhile, smart tourism coupled with sustainable transformation has also led to the innovative use of technology for the quality of life in good governance and the tourism industry [48]. Similarly, in the context of Dunhuang’s smart tourism exploration, the open innovative business mode has been manifested in the diverse smart system, i.e., digitalization management of the tourism market, intelligent operation of international events, intelligent marketing of government official websites, intelligent sale of cultural and creative products and intelligent service/hospitality of tourism attraction, especially the International Dunhuang Project (IDP) conducted by Dunhuang Academy with the British Library about international standard digitization works of Dunhuang scriptures and cultural relics. It could be generalized that Dunhuang’s practice is an important supplement and attempt to explore diversified innovative business models worldwide.

There still exist some limitations of this work. (1) The interacted relations and changes of international tourism and social-ecological sustainability has not been explored in detail in this study, from quantitative measurement to qualitative elaboration. (2) The research lacks in-depth analysis to explore the social equity issues triggered by the development of urban globalization, summarily comprising social justice, social, and social welfare. (3) The eloquent conceptualization and theorization of Dunhuang’s open innovation business model is not well summarized within discourse of China’s socialist political-economic specificity. For the future, additional studies will be needed to address the above gaps. (1) Environmentally, the author plans to supplement the well-rounded consideration on sustainable development. Of vital concern is the interacting relationships, trigger process, evolutionary dynamics amongst international tourism, and social and environmental sustainability through an empirical insight into how sustainable livelihood and ecology feedback positively and negatively to urban globalized transformation via international tourism. (2) Socially, some extensions to social responsibility and diversity triggered by tourism warrant further observation, with a tight focus on the structural change and differentiation of social classes, enhanced participation of people in all walks of life, and then emergence of social diversity. During the process, we should highlight the role of enterprises by which to undertake social responsibility and improve social welfare not confining to the role of government. (3) Creatively, as for a shared innovative business mode, we should strengthen comparative analysis of Dunhuang’s smart tourism and similar practices implemented in the world. Notably, the study necessitates refining the novelty of Dunhuang’s smart tourism model, highlighting the innovation and contribution of Dunhuang’s smart tourism to the shared business model. Moreover, it is worth examining residents’ behavioral mobility and travel based on the quantification of big data sharing.

## Figures and Tables

**Figure 1 ijerph-19-11241-f001:**
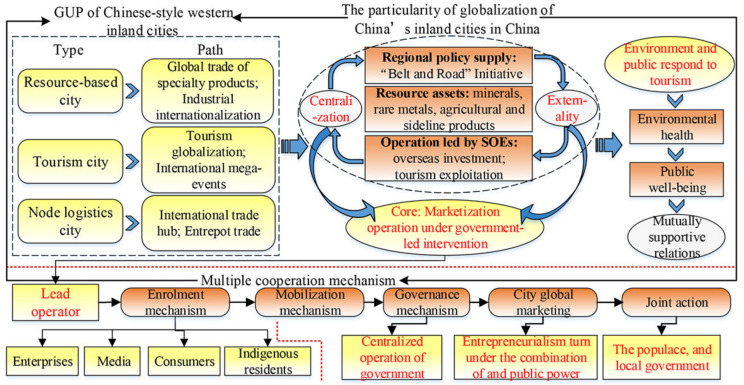
The conceptual framework of UGP of western inland cities in China.

**Figure 2 ijerph-19-11241-f002:**
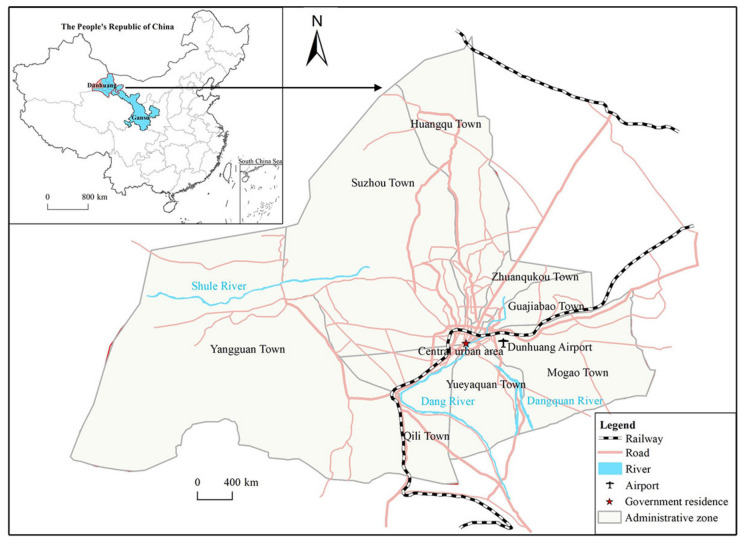
The location of Dunhuang City.

**Figure 3 ijerph-19-11241-f003:**
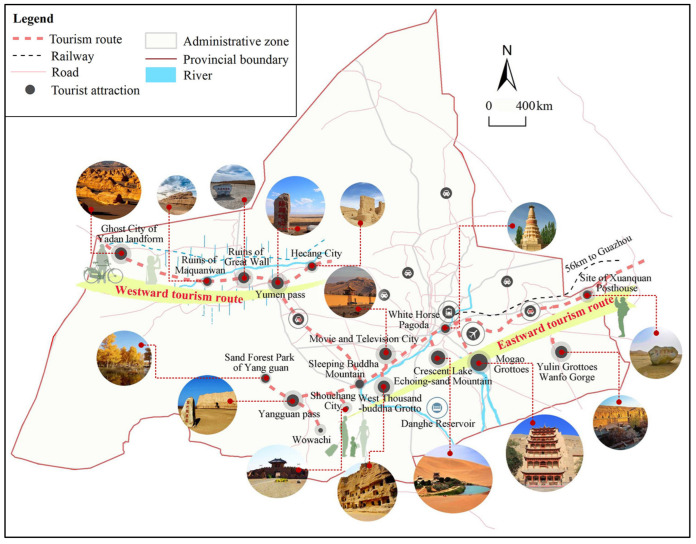
The spatial distribution of tourism resources and routes in Dunhuang.

**Figure 4 ijerph-19-11241-f004:**
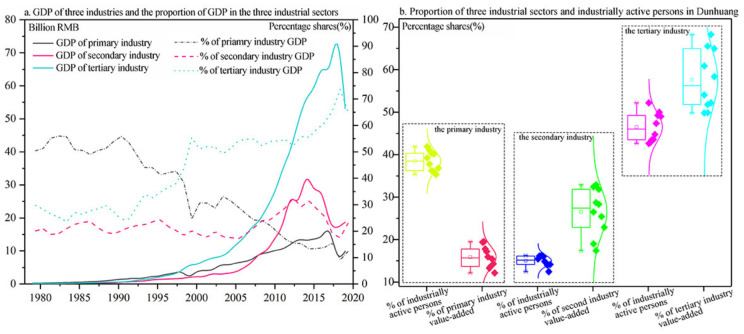
The industrial structure and transformation in Dunhuang.

**Figure 5 ijerph-19-11241-f005:**
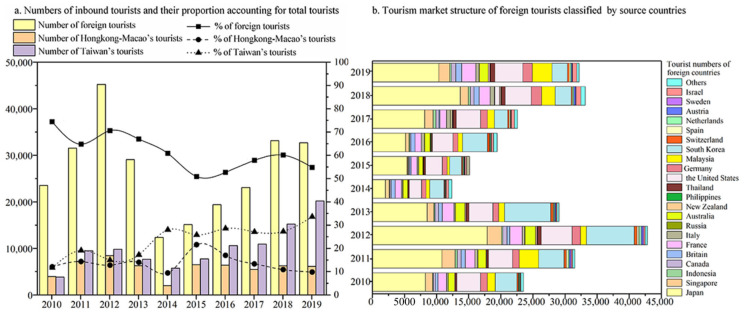
Tourism market structure of numbers and proportions of inbound tourists, 2010–2019.

**Figure 6 ijerph-19-11241-f006:**
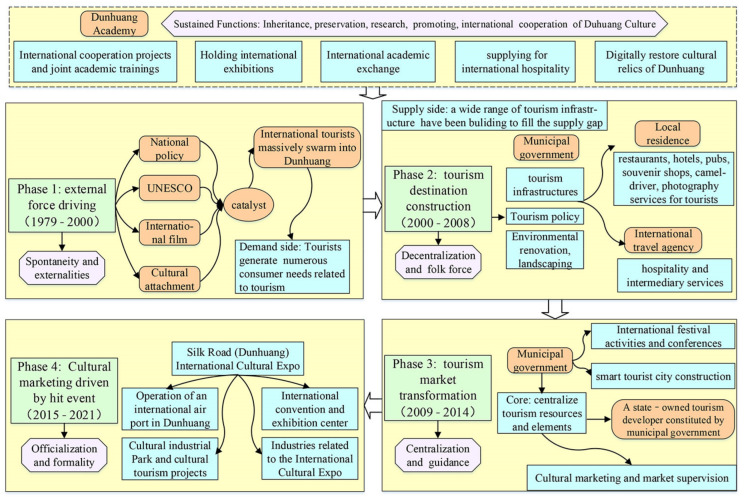
The globalization path and progress of Dunhuang.

**Figure 7 ijerph-19-11241-f007:**
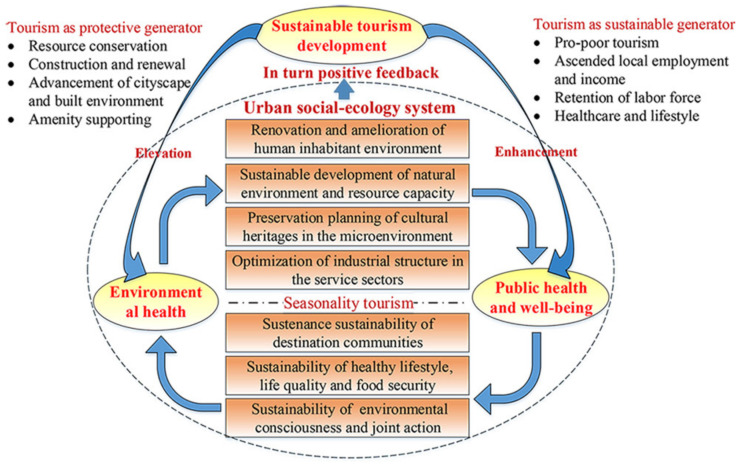
Synergistic relationships between Dunhuang’s tourism and social-ecology system.

**Figure 8 ijerph-19-11241-f008:**
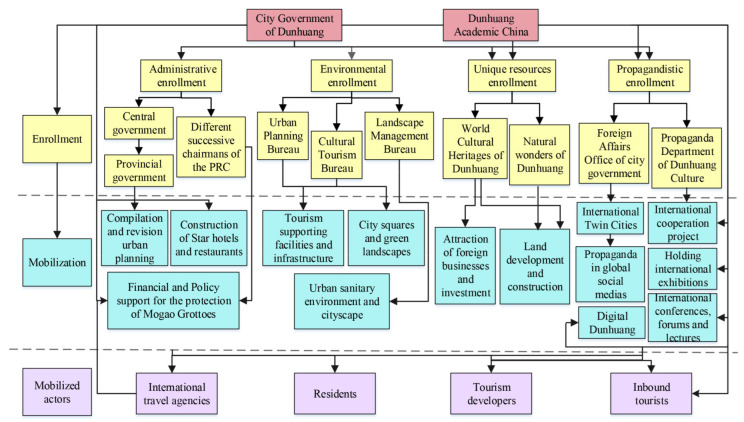
The formation of multiple cooperation network in the phase of world cultural heritage destination construction.

**Figure 9 ijerph-19-11241-f009:**
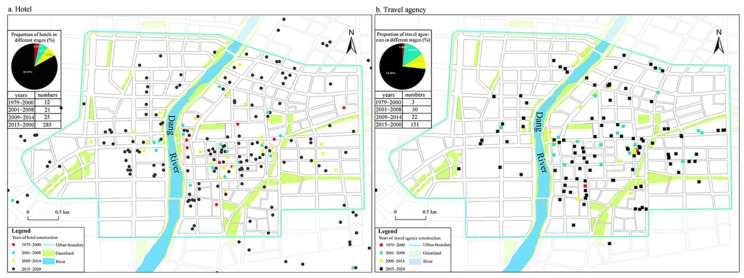
The spatial distribution of hotels and travel agencies in the four evolution phrases.

**Figure 10 ijerph-19-11241-f010:**
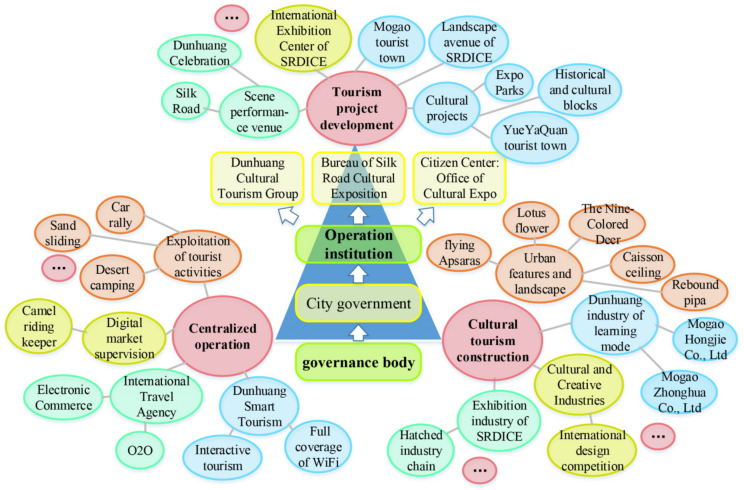
Governance mechanism in the centralized operation and global city marketing phase.

**Figure 11 ijerph-19-11241-f011:**
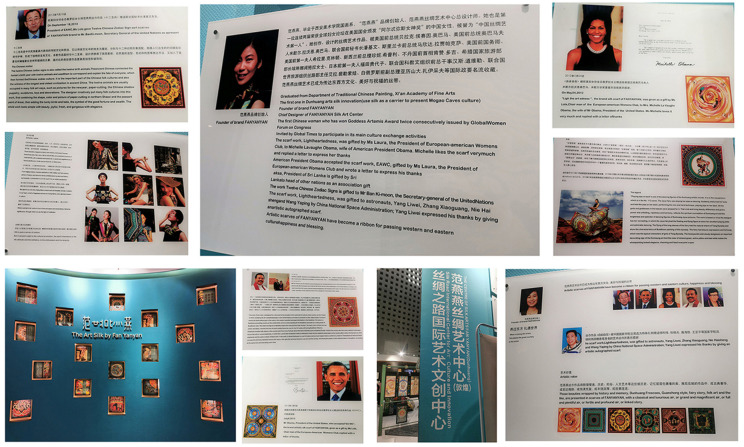
The center of silk arts and a bridge for global cultural communication (Dunhuang).

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
