# Peer review of "The Multiple Cooperative Mechanism and Globalization Path of Small Inland Cities in China: A Showcase Study of Dunhuang, China"

_ijerph, 2022, doi:10.3390/ijerph191811241_

Round 1
Reviewer 1 Report
The submitted article is very well written, with an extensive conceptual and methodological section explaining sufficiently the research design of this article. The results are well presented with all visual material (figures, graphs and maps) needed for illustrating the results, following in some very interesting findings about the urban globalisation paths of Small Inland Cities in China.
However, what is crucially missing from the article, is its relevance with the environmental health and/or public health, which are the main focus areas of this journal (as mentioned in the journal’s page https://www.mdpi.com/journal/ijerph/about). This connection should be made apparent throughout the article, to be suitable for publishing in this journal, and this constitutes the single major revision that is needed in your article. All other (minor) issues are included as comments, in the attached pdf file

Author Response
Response to Reviewer Comments
Dear Reviewer:
We thank you very much for giving us an opportunity to revise our manuscript. Meanwhile, we are also grateful for your positive and constructive insights and comments on manuscript entitled “The multiple cooperative mechanism and globalization path of small inland cities in China: A showcase study of Dunhuang, China” (ID: ijerph-1861268). We have studied reviewer’s comments and have made revision. The following is the concrete revisions:
The submitted article is very well written, with an extensive conceptual and methodological section explaining sufficiently the research design of this article. The results are well presented with all visual material (figures, graphs and maps) needed for illustrating the results, following in some very interesting findings about the urban globalisation paths of Small Inland Cities in China.
Point: However, what is crucially missing from the article, is its relevance with the environmental health and/or public health, which are the main focus areas of this journal (as mentioned in the journal’s page https://www.mdpi.com/journal/ijerph/about). This connection should be made apparent throughout the article, to be suitable for publishing in this journal, and this constitutes the single major revision that is needed in your article. All other (minor) issues are included as comments, in the attached pdf file.
Response: Thank for your pertinent suggestions. I’ve strove to make the connection between the paper and environmental health and public well-being in the abstract, introduction, newly-supplementary analytical section, conclusion, discussion, future work and research limitations, supplementing related contents. In the context of this article, we take the extended meaning of public health as public well-being, environmental health could be interpreted as environment and resource protection, urban built environment and landscape optimization and sustainable tourism development. The term of environmental health and public health can be generalized as urban social-ecology system. I’ve highlighted the relevance in yellow background in the response letter and manuscript article for your convenience.
Revised result:
Abstract:
Environmentally, Dunhuang’s tourism internationalization enhances process of the development of sustainable shared mobility industry, and especially,its tourism development and social-ecology system maintain the synergistic relationships which are international tourism promotes urban ecosystem and public welfare and in turn, social-ecological enhancement serve Dunhuang's international tourism well. (Quoted from the revised article “Abstract”, line number 28-31)
- Introduction:
Dunhuang’s local experience could fertilize other cities on two points: first, it heuristically notes that how to internationalize local unique tourism resources, and further shape the city as an international tourism destination. Eventually, the city’s physical built environment, social-ecological system, public well-being, sustainable environmental awareness and action, service facilities and landscape and greenery designs appear globalizing turn. More essentially, Dunhuang’s experience is the epitome of fully utilizing nationalization power and market mechanism. Therefore, we would mainly argue: in the context of marginal location and insufficient FDI, how an initially enclosed market and economy could break such path lock-ins and integrate into globalization in a small inland city in China? What kinds of cooperation mechanisms between government and related market actors should be established? as well as to what extent tourism and social-ecology system mutually impact and synergistically develop? Concisely, the study further validates and enriches the issue that the locals escalate urban transformation, social-ecology improvement and globalization turn via distinctive resources, but more focuses on the hybrid triggers of "government + SOE + market" under China’s special socialist market economic institution. (Quoted from the revised article “Introduction”, line number 112-115, 120121)
- A conceptual framework of UGP of western inland cities in China
2.1. The urban globalization of western inland cities
More important, the similarities of the above UGP in western China, they may support the relevance of urban globalized transformation and local social-ecology, where environmental and public responsiveness widely exists and intertwined with the local development direction, the social- ecological effects of international tourism in particular (Fig.1). (Quoted from the revised article “2.1 The urban globalization of western inland cities”, line number 164-170)
Figure 1. The conceptual framework of UGP of western inland cities in China.
In the newly-supplementary analytical section
4.2. Tourism-driven urban globalization in Dunhuang
(4) The makeover of cityscape and environment. The mayor S made abundant reformations during his tenure, such as the renovation of cityscape and environment, the makeover of Danghe River Landscape Line, the exploitation of tourism projects, etc.. A government official (Director of development and reform commission, 0929B1) said: “Dunhuang’s urban environment was deadly poor previously. Mayor S considered that such environment was incompatible with an tourist city image. Dunhuang was deemed a world-famous tourist resort, and antiseptic city environment and civilized behavior of the populace should be developed as a mirror of its tourism reputation. So he carried out voluntary cleaning exercise of urban garbage, the sewage treatment and Danghe river landscape construction, and appealed to the public improving tourism service awareness.” Another local resident (1006A5) said: “There existed a key time node in 2005~2006, cityscape and environment appeared dramatic amelioration. Dunhuang was not always a tourism city that only relies on Mogao Grottoes to attract tourists since then. Mayor S said that our livelihood were directly linked to Dunhuang tourism, only when we did well in urban sanitation and landscape can Dunhuang attract ever-increasing tourist arrivals. Hence, everyone was enthusiasm and took care of the cityscape as a habit.” (Quoted from the revised article “ 4.2. Tourism-driven urban globalization in Dunhuang”, stage II, line number 406-422)
4.3 Tourism impacts on social-ecological effects in Duhuang
Benefits in urban social-ecology system induced by tourism are assessed as illustrated in Fig.7. Dunhuang’s tourism development and social-ecology system maintain the synergistic relationships, whose intersection is a two-way interplay process. For one front, international tourism impacts urban ecosystem and public welfare. Environmentally, tourism development have elevated following four aspects, the renovation and amelioration of human inhabitant environment (i.e., cityscape, built environment, greenery, amenity), the sustainable development of natural environment and resource capacity, the preservation planning of cultural heritages (Mogao Grotto in particular, like cave refurbishment, the monitor of mural diseases and pests and microenvironment in caves, desertification governance), and the optimization of industrial structure (reduction of heavily pollution-creating industries and augmentation of eco-friendly service sector). For the indigenous populace, international tourism have enhanced their public well-being portfolio diversification as major sustainability generator in three pathways, that is, the livelihood sustainability of destination communities (pro-poor tourism, increased local employment and income induced by tourism, the retention of labors), the sustainability of healthy lifestyle, life quality and food security, coupled with sustainability of environmental consciousness and joint action. For another front, in turn, above social-ecological enhancement as one agent serve Dunhuang's international tourism well. Dunhuang’s tourism is inextricably interwoven with urban environmental health and public well-being (Fig.7). (Quoted from the revised article “4.3 Tourism impacts on social-ecological effects in Duhuang”, line number 470-491)
Figure 7. Synergistic relationships between Dunhuang’s tourism and social-ecology system
- Conclusions:
(3) As a environmental reflection during globalization, it is pointed out that tourism is not an isolated activity but it interacts with local social-ecology system, forming synergistically supportive relation with environmental health and public well-being. Meanwhile, ameliorated social-ecology system could well serve and foster international tourism. (Quoted from the revised article “6. Conclusions”, line number754-759)
Discussion:
The paper explores UGP of Dunhuang, its MCMGP could nurture other cities on the two implications: 1) locals could tap, operate and market potential resources, by which it initially nurtures the globalization process of an industry, and then even diffuses globalizing transformation of entire urban social-ecology system and public well-being, coupled with escalating quality of built environment and public health. Dunhuang mode can be concisely summarized as such an evolution clue, “the unique tourism resources—international tourism destination—the eco-friendly social-ecology responsiveness—a globalizing tourist city. 2) Dunhuang’s experience is the epitome of fully utilizing organic coupling of government power and market mechanism and exists its novelty not only in China but also other countries. Besides, local government nationalizes the CTG derived from an Internet company, and adopts an open innovation business model like global smart tourism, social media, and tourism community promotion, whose indigenous practice also fertilizes some traditional tourist cities or SOEs. Actually, MCMGP’s hybrid drivers of “centralized governance and limited marketization” has been employed in some regions overseas but exist subtly distinctiveness. For instance, in the Sogndal, Norway case of smart town sports tourism integration, Sogndal Football Club and local university as a catalyst are manifestation of free marketization, they focus on Norwegian top league project to promote the commodification of Sogndal combined with local entrepreneur’s efforts via the resources of intelligence systems, like 5G and Internet of things, and then proceed with ties amongst sport, social-ecology, and intelligent town space. Similarly, Dunhuang’s highly centralized tourism governance mode parallels that for Singapore authorities’ one as noted in Chang’s studies. As a representative city-state, Singapore's globalization is characterized by a top-down state tourism governance alongside a state-envisioned “Chineseness”. Via global intelligent tourism, Singapore Tourism Board (STB) retains the ascendancy, where it devises tourism master plans that portray the country as a melting pot of “Western” and “Asian” cultures by communicating local cultures and improves a eco-friendly pathways. Smart tourism, optimization of resources, sustainable development, has fostered the innovative use of technology for the quality of life in environmental governance and the tourism industry. But comprehensively, unlike the vast majority of tourist towns or cities in western countries are predicated on a self-regulated and free-market economy (i.e.,tourism development tends to private capital and developers, and attraction protection mainly via local community). Dunhuang’s diversity is the government entrepreneurialism that government dominants the market-oriented path to undergird tourism operations rather than that of enterprises, the hand of the state and government underpins tourism construction as a result of “the inner workings of essentially state-envisioned institution”. (Quoted from the revised article “6. Conclusions”, para 3, line number 778-780, 795-796, 799-804)
Research Limitation:
Finally, several limitations to the present study should be considered. First, The tone of the work is inclined to rather descriptive elaboration with a tight focus on globalizing trigger mechanism, mainly looking at interviewing records and statistics. Further consolidated observation necessitates some supplementary quantitative underpinnings concerning the issues, i.e., the diverse perception of local residents to urban globalization, the impact of urban construction on the environmental response, resource preservation and public well-beings in Dunhuang. Second, the MCMGP mode of Dunhuang is a special showcase with limited adaptability and distinctive requirements, like unique resource, government centralization and rather overwhelming market power of SOE. Its practice is more applicable to cities with potential tourism assets along the ‘Silk Road’ than all cities or general tourism cities. (Quoted from the revised article “6. Conclusions”, para 4, line number, 814-817, 820-828)
Future Avenue:
In doing so, additional studies will be needed to accommodate future avenue. (1) A global urban transformation should not overlook the “dark side” as spin-offs from international tourism and external factors. These potential existences are as socio-ecologically produced and economically laden, the diverse erosion of built environment and indigenous culture in particular. The adverse incursion and consequent burden to local environment health, public well-beings, resource preservation warrant further examination when gaining insights into this broader progress. (2) Some extensions to the quantitative methods will be incorporated. First extension of the measurement theme will focus on two way process and reaction of man-land relationship during Dunhuang’s globalizing tourism, especially the environmental responsiveness, governance and sustainable tourism development grounded in indicator modelling.The other quantitative analysis may be supplemented by multi-faced global perception of indigenous residents, and will eloquently expound in international tourism prestige, degree of openness, global visions and thoughts of residents, global presence of cultural tourism and competitiveness of tourism assets. (3) Further observation will create a clear picture to the exact role of local small and medium-sized tourism developers, that is, developing an instrument to determine to what extent those private companies played their role not only concentrate on the CTG (SOE), associated with an empirical insight into how two groups build coupling linkage and collaborative and competitive nexus. (Quoted from the revised article “6. Conclusions”, para 4, line number 10-31)

Reviewer 2 Report
The article concerns an interesting topic which is "The multiple cooperative mechanism and globalization path of small inland cities in China: A showcase study of Dunhuang, China". The topic of the article is very timely and fits into the thematic scope of the journal.
From the linguistic point of view, the article is correct. It is very easy to read and clearly arranged.
From the editorial point of view:
1. Please shorten the abstract as it is a bit too extensive.
2. Please improve the quality of Figure 1. It looks as if it was an excerpt from some article.
3. Please correct the references to the literature to make them compliant with the MDPI requirements.
4. Please improve the quality of Figure 6.
5. Please improve the quality of Figure 8.
6. Please correct the formatting of the literature to comply with the MDPI requirements.
From the substantive point of view:
7. There was no chapter in the article that would be devoted to discussing the results together with confronting them with the practices implemented in the world and described by other scientists. It is worth mentioning that in the world with the development of the phenomenon of globalization, the concept of sustainable development, corporate social responsibility and diversity has also developed widely. Moreover, in recent years the issue of creating open business innovations has also developed. Please refer to 'Diversity as an opportunity and challenge of modern organizations in TSL area' and 'Open Innovation Business Model as an Opportunity to Enhance the Development of Sustainable Shared Mobility Industry'.
8. In turn, in the summary, refer to the further work that you are planning on the topic under study.
9. In conculssion please add also an addnotation about the limitation of your research.
Good luck!
Author Response
Response to Reviewer Comments
Dear Reviewer:
We thank you very much for giving us an opportunity to revise our manuscript titled “The multiple cooperative mechanism and globalization path of small inland cities in China: A showcase study of Dunhuang, China” (ID: ijerph-1861268). Meanwhile, we are also very grateful for your insightful comments, which have benefited the quality of the paper and have made a clearer storyline in the interpretation. We have studied reviewer’s comments and have made revision and marked the corresponding revised parts with red color in the revised manuscript. The main modification in the paper and the responds to the reviewer’s comments are as following:
The article concerns an interesting topic which is "The multiple cooperative mechanism and globalization path of small inland cities in China: A showcase study of Dunhuang, China". The topic of the article is very timely and fits into the thematic scope of the journal.
From the linguistic point of view, the article is correct. It is very easy to read and clearly arranged.
From the editorial point of view:
Point 1. Please shorten the abstract as it is a bit too extensive.
Response 1: Thank you for your instructive comments. We’ve narrowed down the length of abstract, ranging from previous 459 words to current 288 words.
Revised result 1:
Abstract: Currently, urbanization driven by global capital flows entails a main trend in many large cites in China, while global capital investment in small inland cities especially in the western China is extremely scarce, where their globalization characters the powerful nationalization power and market activation. Dunhuang, a small inland city in western China, has transformed successfully from an agricultural county to an international tourist city, a platform for worldwide cultural communication, and a node city in the Belt and Road region because of its unique and brilliant resources Mogao Grottoes and Dunhuangology. Therefore, the paper develops a conceptual framework of the multiple cooperative mechanisms and globalization path (MCMGP) of Dunhuang, elaborating the process of industrial transformation, urban globalization, and multiple cooperative mechanisms between government and market actors based on interviewing records and statistics. Findings show that the MCMGP features government-led intervention, resources orientation, and centralization embodied the driver of state-owned enterprises (SOEs). Also, the MCM in Dunhuang's globalization contains the mechanism of enrolment, mobilization and action, governance and global marketing, distributed in the two phases. Equally important, in respond to Belt and Road Initiative (BRI) and Silk Road (Dunhuang) International Cultural Expo (SRDICE) from the State, city government significantly reinvests and refines cultural tourism via governance mechanism, carving out a key node city in the Silk Road and an international tourist city. Dunhuang experience notes that locals could tap, operate and market potential resources, by which it initially nurtures the globalization process of an industry, and then even diffuses globalizing transformation of entire urban system. Also, Chinese inland cities could learn lessons from Dunhuang path exploiting nationalized market operation and global smart tourism, which is a representative localized practice in the China's peculiar socialist market economy discourse. (Quoted from the revised article “abstract”, line number 10-32)
Point 2. Please improve the quality of Figure 1. It looks as if it was an excerpt from some article.
Response 2: Thank you for detailed and kind advise. Figure.1. is drawn by myself, but I’m so sorry for the misunderstanding caused by my too tight and close layout between the last sentence of text and figure.1, likely leading to some text remains when editor conducts some journal-specific standard editing service in pdf. Format (see in Illustration.1). Thus, I’ve adjusted the unsuitable layout in the revised article, and upload its editable drawing file in Visio software to the submission system and official email as the attachment. (see details in the revised article Fig.1, line number 174-175)
Illustration.1 The screenshot in the original article with doc. Format
Original figure re-inserted in revised article with JPG. Format
The screenshot in the original drawing platform
Point 3. Please correct the references to the literature to make them compliant with the MDPI requirements.
Response 3: Thank you for careful suggestion. I’ve modified the literature citation format to fit into the standards of MDPI in the revised article. ( See the full text for revised details)
Point 4. Please improve the quality of Figure 6 and Figure 8.
Response 4: I’m not sure whether the quality of figures here image resolution or not, you mean, low quality pixels and less clear visualization? If so, I’ve optimized the two figure files reaching 600 dpi and re-inserted them into revised article. but the submission system has limited uploading capacity for revised manuscript, lower than 10M. Hence, I send the higher-capacity revised article to the editor board by e-mail.
(Also, due to my the newly-supplementary figure in 4.3. section of revised manuscript, which causesthe figure order change from the original figure.8. to current figure.9.)
Point 5. Please correct the formatting of the literature to comply with the MDPI requirements.
Response 5: Thank for your sincere tip. I’ve standardized the formatting of the literature according to ijerph-template file, to in line with the reference requirements. (see more details in the revised article References, line number 834-902)
From the substantive point of view:
Point 6. There was no chapter in the article that would be devoted to discussing the results together with confronting them with the practices implemented in the world and described by other scientists. It is worth mentioning that in the world with the development of the phenomenon of globalization, the concept of sustainable development, corporate social responsibility and diversity has also developed widely. Moreover, in recent years the issue of creating open business innovations has also developed. Please refer to 'Diversity as an opportunity and challenge of modern organizations in TSL area' and 'Open Innovation Business Model as an Opportunity to Enhance the Development of Sustainable Shared Mobility Industry'.
Response 6: Thank for your informative comment and guidance. I’ve made the comparison between such strong government centralization and intervention couple with innovative business mode of intelligent system in Dunhuang and other regional implementations in the world, highlighting their similarities and differences.
Revision result 8:
Actually, MCMGP’s hybrid drivers of “centralized governance and limited marketization” has been employed in some regions overseas but exist subtly distinctiveness. For instance, in the Sogndal, Norway case of smart town sports tourism integration, Sogndal Football Club and local university as a catalyst are manifestation of free marketism, they focus on Norwegian top league project to promote the commodification of Sogndal combined with local entrepreneur’s efforts via the resources of intelligence systems, like 5G and Internet of things, and then proceed with ties amongst sport, social-ecology, and intelligent town space. Its essence underscores the self-regulation of market mechanism encompassing shared innovative business mode in the private sectors. Similarly, Dunhuang’s highly centralized tourism governance mode parallels that for Singapore authorities’ one as noted in Chang’s studies. As a representative city-state, Singapore's globalization is characterized by a top-down state tourism governance alongside a state-envisioned “Chineseness”. Via global intelligent tourism, Singapore Tourism Board (STB) retains the ascendancy, where it devises tourism master plans that portray the country as a melting pot of “Western” and “Asian” cultures by communicating local cultures and improves a eco-friendly pathways. Smart tourism, optimization of resources, sustainable development, has fostered the innovative use of technology for the quality of life in the versed governance and the tourism industry. But comprehensively, unlike the vast majority of tourist towns or cities in western countries are predicated on a self-regulated and free-market economy (i.e.,tourism development tends to private capital and developers, and attraction protection mainly via local community). Dunhuang’s diversity is the government entrepreneurialism that government dominants the market-oriented path to undergird tourism operations rather than that of enterprises, the hand of the state and government underpins tourism construction as a result of “the inner workings of essentially state-envisioned institution”. (Quoted from the revised article “6. conclusions”, para 2, line number 787-810)
Point 7. In turn, in the summary, refer to the further work that you are planning on the topic under study.
Revision result 7:
In doing so, additional studies will be needed to accommodate future avenue. (1) A global urban transformation should not overlook the “dark side” as spin-offs from international tourism and external factors. These potential existences are as socio-ecologically produced and economically laden, the diverse erosion of built environment and indigenous culture in particular. The adverse incursion and consequent burden to local environment health, public well-beings, resource preservation warrant further examination when gaining insights into this broader progress. (2) Some extensions to the quantitative methods will be incorporated. First extension of the measurement theme will focus on two way process and reaction of man-land relationship during Dunhuang’s globalizing tourism, especially the environmental responsiveness, governance and sustainable tourism development grounded in indicator modelling. The other quantitative analysis may be supplemented by multi-faced global perception of indigenous residents, and will eloquently expound in international tourism prestige, degree of openness, global visions and thoughts of residents, global presence of cultural tourism and competitiveness of tourism assets. (3) Further observation will create a clear picture to the exact role of local small and medium-sized tourism developers, that is, developing an instrument to determine to what extent those private companies played their role not only concentrate on the CTG (SOE), associated with an empirical insight into how two groups build coupling linkage and collaborative and competitive nexus. (Quoted from the revised article “6. conclusions”, para 3, line number 820-836)
Point 8. In conculssion please add also an addnotation about the limitation of your research.
Good luck!
Revision result 8:
Finally, several limitations to the present study should be considered. First, The tone of the work is inclined to rather descriptive elaboration with a tight focus on globalizing trigger mechanism, mainly looking at interviewing records and statistics. Further consolidated observation necessitates some supplementary quantitative underpinnings concerning the issues, i.e., the diverse perception of local residents to urban globalization, the impact of urban construction on the environmental response, resource preservation and public well-beings in Dunhuang. Second, the MCMGP mode of Dunhuang is a special showcase with limited adaptability and distinctive requirements, like unique resource, government centralization and rather overwhelming market power of SOE. Its practice is more applicable to cities with potential tourism assets along the ‘Silk Road’ than all cities or general tourism cities, and thus the results finitely feed into others. (Quoted from the revised article “6. conclusions”, para 3, line number 812-820)

Round 2
Reviewer 1 Report
Dear authors,
I appreciate your effort for trying (and succeeding) in the critical -in my view- issue of your article: its relevance with the environmental and/or public health. Therefore, I do not have major comments for your article.
However, I would advise you to consider the minor issues that I have pointed out, even from the previous stage, included as comments in the attached pdf file. As I can see you have not incorporated them in the revised version of your article.

Author Response
Response to Reviewer 1 Comments
Dear Reviewer:
We thank you very much for giving us an opportunity to revise our manuscript. Meanwhile, we are also grateful for your positive and constructive insights and comments on manuscript entitled “The multiple cooperative mechanism and globalization path of small inland cities in China: A showcase study of Dunhuang, China” (ID: ijerph-1861268). We have studied reviewer’s comments and have made revision. The following is the concrete revisions:
I appreciate your effort for trying (and succeeding) in the critical -in my view- issue of your article: its relevance with the environmental and/or public health. Therefore, I do not have major comments for your article.
Point: However, I would advise you to consider the minor issues that I have pointed out, even from the previous stage, included as comments in the attached pdf file. As I can see you have not incorporated them in the revised version of your article.
Response: Thank you for your careful advise and sincere encouragement. I’m so sorry for the incompatibility issue of my PDF. Reader, it’s unavailable for me to see some notes in the attached pdf file in the last stage, like the following screenshot. I’ve downloaded related word software where these comments is available to me this time. I’m so sorry again.
Note1: Reference the research (articles, studies etc) that you are referring to.
Revised result 1: After revisiting previous research [3,6-12,17], (Quoted from the revised article “Introduction”, para 3, line number 78)
Note2: You should explain what this abbreviation means (I understand that it means "Urban Globalization Path", but it should be made clearer for the reader).
Revised result 2: Unlike urban globalization path (UGP) of western countries or eastern coastal China within above such contexts, (Quoted from the revised article “Introduction”, para 3, line number 82)
Note3: Reference the research (articles, studies etc) that you are referring to. Revised result 3: However, little research paid attention to such leading role of resources’ innate triggering factor and government-led multiple synergy [38,40-41],
(Quoted from the revised article “Introduction”, para 3, line number 87)
Note4: You should avoid abbreviations in your section title.
Revised result 4: 2. A conceptual framework of urban globalization path of western cities in China. (Quoted from the revised article “2. section title”, line number 127)
Note5: This figure is very usefull for understanding the conceptual framework of UGP of western inland cities in China.
To this end, it should be included in higher resolution
Revised result 5: I’ve optimized the two figure files reaching 600 dpi and re-inserted them into revised article. but the submission system has limited uploading capacity for revised manuscript, lower than 10M. Hence, I send the higher-capacity revised article to the editor board by e-mail.
Figure 1. The conceptual framework of UGP of western inland cities in China.
Note6: You should avoid abbreviations in your section title.
Revised result 6: 5.3.3. The functions of different-level government and the characteristics of Dunhuang tourism-oriented state-owned enterprise (Quoted from the revised article “2. section title”, line number 704-705)

Reviewer 2 Report
Dear authors, thank you for your corrections. I'm sorry, but unfortunately you still haven't fully addressed them. The article is still not supplemented with important substantive issues that I mentioned previously. Please fill in the indicated deficiencies:
7. There was no chapter in the article that would be devoted to discussing the results together with confronting them with the practices implemented in the world and described by other scientists. It is worth mentioning that in the world with the development of the phenomenon of globalization, the concept of sustainable development, corporate social responsibility and diversity has also developed widely. Moreover, in recent years the issue of creating open business innovations has also developed. Please refer to 'Diversity as an opportunity and challenge of modern organizations in TSL area' and 'Open Innovation Business Model as an Opportunity to Enhance the Development of Sustainable Shared Mobility Industry'.
8. In turn, in the summary, refer to the further work that you are planning on the topic under study.
9. In conculssion please add also an addnotation about the limitation of your research.
Author Response
Response to Reviewer 2 Comments
Dear Reviewer:
Thank you for your comments concerning our manuscript entitled “The multiple cooperative mechanism and globalization path of small inland cities in China: A showcase study of Dunhuang, China” (ID: ijerph-1861268). Your comments are very valuable and very helpful for revising and improving our paper, as well as the important guiding significance to our researchers. We have studied comments carefully and have made correction which we hope meet with approval. The main corrections in the paper and the responds to the reviewer’s comments are as following.
Dear authors, thank you for your corrections. I'm sorry, but unfortunately you still haven't fully addressed them. The article is still not supplemented with important substantive issues that I mentioned previously. Please fill in the indicated deficiencies:
Point 7: There was no chapter in the article that would be devoted to discussing the results together with confronting them with the practices implemented in the world and described by other scientists. It is worth mentioning that in the world with the development of the phenomenon of globalization, the concept of sustainable development, corporate social responsibility and diversity has also developed widely. Moreover, in recent years the issue of creating open business innovations has also developed. Please refer to 'Diversity as an opportunity and challenge of modern organizations in TSL area' and 'Open Innovation Business Model as an Opportunity to Enhance the Development of Sustainable Shared Mobility Industry'.
Response 7: Thank you for giving us the precious revision opportunity. We have rewritten the revised result supplementing analysis with related references. We’ve made our earnest endeavor to revise the three questions again, so as to be accordance with your expected results. But, considering the limited length of journal article, we choose the most relevant ones to conduct comments as detailed as possible, in order to guarantee the article’s appropriate length.
Revision result 7:
- International tourism is supportive tourban sustainable development transformation associated with enhancement of residents’ sustainable livelihoods [43]. Nunkoo argued that tourism is an eco-friendly and smokeless industry relative to traditional economic ones[44]. It is also beneficial to facilitate the industrial structure optimization as consequence of replacement of heavy-polluted industry[45]. More importantly, it is useful and practical to guide exploration of the contribution of tourism to residents’ livelihoods via diverse employment and income availability[43]. Overall, Dunhuang’s international tourism and social ecosystem have also formed a virtuous circle of human-land relationship, further have fostered social and environmental justice. 2) Dunhuang’s international tourism promotes urban diversified system, which pertains to a microcosm of urban globalized transition. In Golba’s and TuroÅ„’s productive works, diversity management has spread widely, which more and more attention to attach organizations in all industries, Transport-Shipping-Logistics in particular[46,47]. Similarly,Dunhuang’s international tourism has also diffused to all walks of life in this work, mapping multi-faced social capitals into the synergistic diversity ecosystem. Concisely, employment diversification, business diversification, industrial chain diversification, cityscape diversification, culture stock diversification, governance and operation diversification, residential lifestyle diversification, etc.. 3) Dunhuang’s smart tourism is one of most vivid representatives open innovation business modes. In effect, from the angle of open shared mobility and sustainable transport system, the issue of creating open business innovations has substantially developed in TuroÅ„’s seminal research[47]. Meanwhile, smart tourism coupled with sustainable transformation, has aso led to the innovative use of technology for the quality of life in good governance and the tourism industry [48]. In a similar vein, in the contexts of Dunhuang’s smart tourism exploration, the open innovative business mode has been manifested the diverse smart system, e., digitalization management of tourism market, intelligent operation of international events, intelligent marketing of government official websites, intelligent sale of cultural and creative products and intelligent service / hospitality of tourism attraction, especially the International Dunhuang Project (IDP) conducted by Dunhuang Academy China with the British Library about international standard digitization works of Dunhuang scriptures and cultural relics. It could be generalized that Dunhuang’s practice is an important supplement and attempt to explore diversified innovative business models in the world.
Supplemented References
43.Su M.; Wall G.; Wang Y.; Jin, M. Livelihood sustainability in a rural tourism destination - Hetu Town, Anhui Province, China. Tourism Management. 2019, 71(4.): 272-281.
44.Nunkoo, R. Governance and Sustainable Tourism: What is the Role of Trust, Power and Social Capital?. Journal of Destination Marketing and Management. 2017, 6, 277-285.
45.Sun J.; Chen J., Huang X. The bargain between subjects and rights negotiation in the tourism environmental governance issue of Erhai in Dali. Scientia Geographica Sinica, 2020, 40(9):1468-1475.
46.Golba, D., Turoń, K., Czech, P. Diversity as an opportunity and challenge of modern organizations in TSL area. Scientific Journal of Silesian University of Technology. Series Transport. 2016, 90, 63-69.
47.Turo’n, K. Open Innovation Business Model as an Opportunity to Enhance the Development of Sustainable Shared Mobility Industry. J. Open Innov. Technol. Mark. Complex. 2022, 8, 37.
48.Liao C.; Nong L. Smart City Sports Tourism Integration Based on 5G Network and Internet of Things[J]. Microprocessors and Microsystems. 2021, 6, 1-16.
Point 8: In turn, in the summary, refer to the further work that you are planning on the topic under study.
Revision result 8:
Future avenue :
For the future, additional studies will be needed to address above gaps. 1) Environmentally, the author plans to supplement the well-round consideration on sustainable development. Of vital concern is interacted relationship, trigger process, evolutionary dynamics amongst international tourism, social and environmental sustainability, through an empirical insight into how sustainable livelihood and ecology feedback positively and negatively to urban globalized transformation via international tourism; 2) Socially, some extensions to social responsibility and diversity trigger by tourism warrant further observation, with a tight focus on the structural change and differentiation of social classes, enhanced participation of people in all walks of life, and then emergence of social diversity. During the process, we should highlight the role of enterprises by which to undertake social responsibility and improve social welfare not confining to the role of government; 3) Creatively, as for shared innovative business mode, we should strengthen comparative analysis of Dunhuang’s smart tourism and similar practices implemented in the world. Notably, the study necessitates refining the novelty of Dunhuang’s smart tourism model, highlighting the innovation and contribution of Dunhuang’s smart tourism to the shared business model. Moreover, it is worth examining residents’ behavioral mobility and travel based on the quantification of big data sharing.
Point 9: In conculssion please add also an addnotation about the limitation of your research.
Revision result 9:
Limitation:
Finally, there still exist some limitation of this work. 1) The interacted relations and changes of international tourism and social-ecological sustainability has not been explored in detail in this study, from quantitative measurement to qualitative elaboration; 2) The research lacks in-depth analysis to explore the social equity issues
triggered by the development of urban globalization, summarily comprising social justice, social sharing and social welfare. 3) The eloquent conceptualization and theorization of Dunhuang’s open innovation business model, is not well summarized within discourse of China’s socialist political-economic specificity.
